

# Evidence for microbial mediated nitrate cycling within floodplain sediments during groundwater fluctuations

Nicholas J Bouskill[1], Mark E Conrad[1], Markus Bill[1], Eoin L Brodie[1], Yiwei Cheng[1], Chad Hobson[1], Matthew Forbes[2], Karen L Casciotti[2], and Kenneth H Williams[1]

[1]Earth and Environmental Sciences Area, Lawrence Berkeley National Laboratory, Berkeley, CA, 94720
[2]Department of Environmental Earth System Science, Stanford University, Stanford, CA, 94305

*Correspondence to:* Nicholas Bouskill (njbouskill@lbl.gov)

**Abstract.** The capillary fringe is a subsurface terrestrial-aquatic interface which can be a significant hotspot for biogeochemical cycling of terrestrially derived organic matter and nutrients. However, pathways of nitrogen (N) cycling within this environment are poorly understood, and observations of temporally discrete changes in nitrate concentrations lack the necessary resolution to partition between biotic or abiotic mechanisms. Here we take an experimental and mechanistic modeling approach to characterize the annual decline of nitrate ($NO_3^-$) within floodplain sediments at Rifle, Colorado. At discrete sampling points during 2014 we measured $NO_3^-$, ammonia ($NH_4^+$), gaseous nitrous oxide ($N_2O$) and the corresponding isotopic composition of $NO_3^-$. Coincident with an annual spring/ summer excursion in groundwater elevation driven by snowmelt, we observed a rapid decline in $NO_3^-$ concentrations at three depths (2, 2.5 and 3 m) below the ground surface. Isotopic measurements (i.e., $\delta^{15}N$ and $\delta^{18}O$ of $NO_3^-$) suggest an immediate onset of biological N loss at 2 m. At 2.5 and 3 m, $NO_3^-$ concentrations declined initially with no observable isotopic response, indicating an initial dilution of $NO_3^-$ within the well. Following extended saturation by groundwater at these depths we observed subsequent nitrate reduction. A simple Rayleigh model suggests depth-dependent variability in the importance of actively fractionating mechanisms (i.e., nitrate reduction) relative to non-fractionating mechanisms (mixing and dilution). Nitrate reduction was calculated to be responsible for 64 % of the $NO_3^-$ decline at 2 m, 28 % at 2.5 and 47 % at 3 m, respectively. Furthermore, we observed the highest concentrations of $N_2O$ as groundwater saturated the 2 and 2.5 m depth, concomitant with enrichment of the $\delta^{15}N_{NO3}$ and $\delta^{18}O_{NO3}$. A mechanistic microbial model representing the diverse physiology of nitrifiers, facultative aerobes (including denitrifiers), and anammox bacteria indicates that the bulk of biological N loss within the capillary fringe is attributable to denitrifying heterotrophs. However, this relationship is dependent on the coupling between aerobic and anaerobic microbial guilds at the oxic-anoxic interface. Modeling insights also suggest that anammox might play a more prominent role in N loss under conditions where organic matter concentrations are low and rapidly depleted by aerobic heterotrophs prior to the rise of the water table.

## 1 Introduction

Subsurface terrestrial-aquatic interfaces are hotspots for biogeochemical cycling of organic matter and nutrients (McClain et al., 2003; Lohse et al., 2009), and particularly nitrogen (Hefting et al., 2004; Heffernan et al., 2012; Zhu et al., 2013; Gonnea



and Charette, 2014; Smith et al., 2015). Seasonal event-driven fluctuations in water table height can alter trace gas dynamics (Haberer et al., 2012), substrate availability (Persson et al., 2015), and the distribution of microbial metabolisms (Berkowitz et al., 2004). These activities can promote the formation of sharp oxic/anoxic gradients, which facilitate the spatial and temporal coupling of aerobic and anaerobic metabolisms. Previous work characterizing the subsurface nitrogen cycle has observed the

accumulation and dissipation of $NO_3^-$ at the capillary fringe concomitant with the rise and fall of the water table (Hefting et al., 2004; Abit et al., 2008; Sorensen et al., 2015). However, the mechanistic basis for subsurface $NO_3^-$ dynamics are still largely unknown beyond the inference of several interacting and interdependent abiotic and biotic processes. As the incursion of anthropogenic $NO_3^-$ into groundwater and aquifers continues over the next century it becomes increasingly important to distinguish between potential pathways determining the fate of $NO_3^-$.

Herein, we examine seasonal $NO_3^-$ dynamics around the capillary fringe in an alluvial floodplain at a field site in Rifle, Colorado, USA. Above the water table near-atmospheric concentrations of oxygen provide a niche for diverse groups of aerobic and facultative metabolisms responsible for the turnover of specific carbon pools (Stegen et al., 2016) and the release of $NH_4^+$ that can be nitrified to $NO_3^-$ (Smith et al., 2006). Nitrification is a two-step process. The rate-limiting first step, the oxidation of $NH_4^+$ to $NO_2^-$ via a hydroxylamine ($NH_2OH$) intermediate, is carried out by obligately aerobic chemolithoautotrophic

bacteria and archaea (Ward, 2011), that have previously observed to be present in high abundance within the vadose zone (Hug et al., 2015b; Anantharaman et al., 2016). $NO_2^-$ can be subsequently oxidized to $NO_3^-$ via a diverse group of bacteria, some of which are mixotrophic (Le Roux et al., 2016). This coupling of organic matter mineralization and nitrification under oxic conditions can lead to the accumulation of high concentrations of $NO_3^-$, which can be augmented by atmospheric deposition and infiltration of $NO_3^-$ into the vadose zone (Einsiedl and Mayer, 2006).

On the other hand, spring snowmelt leads to a rise water table height through the capillary fringe and into the vadose zone. This action can alter capillary fringe redox through reduced gaseous exchange with the unsaturated zone, and reduction of oxygen availability (Haberer et al., 2012; Jost et al., 2015). This, in turn, can select for different microbial traits (Hug et al., 2015a; Anantharaman et al., 2016) leading to the elevated activity of anaerobic metabolisms (Heffernan et al., 2012; Zhu et al., 2013). Such event driven changes in geochemical conditions have previously been observed to accompany a decline in $NO_3^-$

concentrations (Abit et al., 2008), which could reflect biological $NO_3^-$ reduction preceding gaseous nitrogen loss (as either $N_2$ or $N_2O$). Metabolisms, including denitrification ($NO_3^- \rightarrow NO_2^- \rightarrow NO \rightarrow N_2O \rightarrow N_2$), by facultative aerobic heterotrophic bacteria, $NO_3^-$-dependent sulfide and iron oxidation and anammox, the anaerobic oxidation of $NH_4^+$ to $N_2$ using $NO_2^-$ as an electron acceptor, have all been shown to occur within the Rifle floodplain (Hug et al., 2015a; Anantharaman et al., 2016; Jewell et al., 2016). Anammox has previously been reported to be of widespread importance to nitrogen loss within aquifers

and groundwater (Erler et al., 2008; Robertson et al., 2011; Smith et al., 2015), with contributions to nitrogen loss rivaling denitrification in some cases (Zhu et al., 2013; Smith et al., 2015). On the other hand, solute dilution during groundwater rise is a plausible explanation for the a drop in measureable $NO_3^-$ concentration, with little loss of $NO_3^-$ from the aquifer. Indeed, previous geochemical characterization of the Rifle groundwater show little to no dissolved oxygen (DO) or $NO_3^-$ (Zachara et al., 2013). However, oxygen trapped in pore spaces can become entrained in the groundwater as it rises into the unsaturated



zone (Yabusaki et al., 2017). This oxygenation would reduce the likelihood of anaerobic metabolic activity during inundation (at least briefly), yet still lead to a drop in $NO_3^-$ concentrations.

The objectives of the present manuscript are two-fold: (1) to characterize the nitrogen biogeochemistry of the Rifle sub-surface as snowmelt driven fluctuations in water table depth change the saturation profile of the vadose zone soils, and (2)

to parse out the biological metabolisms principally responsible for nitrogen cycling during this period. At discrete sampling points over one year we measured different inorganic nitrogen species from pore water samples ($NO_3^-$, $NO_2^-$, $NH_4^+$), and gaseous measurements of nitrous oxide concentrations. Furthermore, we measured the corresponding isotopic composition, $\delta^{15}N$ and $\delta^{18}O$, of $NO_3^-$. The stable isotopes of $NO_3^-$ are an ideal tool for addressing our initial question concerning the contribution of different pathways to the formation and loss of $NO_3^-$ as the water table rises and falls. The characteristic

fractionation of $\delta^{15}N$ and $\delta^{18}O$, of $NO_3^-$ associated with bacterial nitrate reduction (Granger et al., 2008) has previously been used to identify biological activity in subsurface environments (Böhlke et al., 2006a; Kendall et al., 2007; Frey et al., 2014; Clague et al., 2015). Over the course of $NO_3^-$ reduction via denitrification, active fractionation of $\delta^{15}N$ and $\delta^{18}O$ of $NO_3^-$ has been shown to enrich the residual $NO_3^-$ pool between + 5 and +25 ‰ (Granger and Wankel, 2016). Conversely, as the groundwater at Rifle tends to be $NO_3^-$-deficient (Zachara et al., 2013), rising water-table height could dilute capillary fringe

$NO_3^-$ concentrations. Because dilution imparts no isotopic fractionation on the $NO_3^-$ pool, it is possible to broadly separate abiotic and biotic mechanisms by measuring shifts in $\delta^{15}N$ and $\delta^{18}O$ of $NO_3^-$.

Finally, in an effort to parse out the microbial mechanisms contributing to the cycling of $NO_3^-$ and different intermediates of the nitrogen cycle across tight redox gradients we will parameterize and simulate the microbial ecology of the capillary fringe using a trait-based model developed on the basis of previous molecular-based surveys of the microbial community in

the Rifle subsurface (Hug et al., 2015a; Anantharaman et al., 2016; Jewell et al., 2016). Trait-based models are relevant tools for this objective as they simplify the range of microbial diversity within a functional guild by parameterizing each group with a unique combination of trait values governing an organism's fitness under dynamic environmental conditions (Follows et al., 2007; Allison, 2012). This model can therefore be used to interpret observations from the current study and predict outcomes beyond the temporal and spatial limits of field observations.

## 2    Materials and methods

### 2.1    Field Site Description

The Rifle field site is a small ($\sim$ 9 ha) floodplain lying adjacent to the Colorado River. The site hosted a former uranium mill processing facility, and has been intensely studied in the decades since closure (Yabusaki et al., 2007; Williams et al., 2011; Hug et al., 2015a). The unconfined aquifer under the floodplain is composed of unconsolidated sands, silts, clays and gravel

deposited by the river, sitting atop a relatively impermeable layer of the Miocene Wasatch formation (Williams et al., 2011). The groundwater typically lies 3.5 m below ground surface (bgs), but fluctuates annually. During snowmelt (starting in April), the groundwater table can rise by 1 - 1.5 m into the unsaturated zone. This groundwater rise can persist for a period of weeks, with apparent interannual variability related to the magnitude of discharge in the Colorado River adjoining the site on its southern





boundary. The TT03 monitoring location at the Rifle field site consists of a series of vertically resolved suction lysimeters and gas sampler ports installed at the following depths: 0.5, 1.0, 1.5, 2.0, 2.5, 3.0, and 3.14 meters below surface depth (bsd). Installation and well sampling have recently been described in full (Tokunaga et al., 2016). Briefly, lysimeters and gas sampler ports were installed as part of a drilling operation from the ground surface to the water table ($\sim$ 3.25m at the time of drilling).

Porous ceramic cup lysimeters (Soilmoisture Equipment, Corp.; Tucson, AZ) were placed at the aforementioned depths. A groundwater monitoring well adjacent to the vertically resolved lysimeters was used to track variations in groundwater quality parameters (including temperature, pH, specific conductivity, oxidation reduction potential, and dissolved oxygen). Samples for the present study were taken between March and September of 2014, a time period during which measurements from previous years have shown a decline in $NO_3^-$ during groundwater incursion into the unsaturated zone. Fluids recovered from each

lysimeter were filtered (0.45 $\mu$M) and immediately analyzed for $NO_3^-$ and $NO_2^-$ concentrations via anion chromatography (Dionex, Corp. ICS-2100, Sunnyvale, CA) using an AS-18 anion exclusion column. Field measurements of $NO_3^-$ were further corroborated in the laboratory by colorimetric reduction of $NO_3^-$ to $NO_2^-$ via vanadium(III) chloride through a previously described protocol (Bouskill et al., 2013). Porewater $NH_4^+$ concentrations were also measured colorimetrically via reduction by sodium salicylate (Allison et al., 2008).

**2.2   Dual-isotope measurements**

The isotope ratios of $NO_3^-$ ($\delta^{15}N_{NO3}$ and $\delta^{18}O_{NO3}$), where $\delta(‰) = (R_{NO_3}/R_{std} - 1)*1000$, R indicates either $^{15}N/^{14}N$ or $^{18}O/^{16}O$ and 'std' refers to a standard reference material, either $N_2$ in air for $\delta^{15}N$ or Vienna standard mean ocean water (VSMOW) for $\delta^{18}O$, were measured by the denitrifier method (Sigman et al., 2001; Casciotti et al., 2002). $NO_2^-$, which interferes with the analysis, was initially removed from the porewater samples prior to analysis using sulfamic acid, according to

a previously published method (Granger and Sigman, 2009). Water samples were injected into a suspension of *Pseudomonas aureofaciens*, which lacks the $N_2O$ reductase, and quantitatively converts the $NO_3^-$ to $N_2O$. The $N_2O$ was analyzed on Finnigan Delta$^{PLUS}$ XP isotope ratio mass spectrometer connected to a Finnigan GasBench. Individual $N_2O$ injection samples were standardized by comparison to $NO_3^-$ isotope standards USGS32, USGS34, and USGS35 (Böhlke et al., 2003). Samples were measured in triplicate and the $\delta^{15}N$ and $\delta^{18}O$ of $NO_3^-$ reported using the notation ‰ relative to atmospheric $N_2$ and

VSMOW, respectively. $^{15}N_{NO3}$ isotope fractionation can be calculated by a simple Rayleigh fractionation equation, where the $\delta^{15}N$ of the reactant is dependent on the isotope fractionation factor ($\epsilon$), the fraction of initial reactant remaining ($f$), and the $\delta^{15}N$ of the initial reactant, according to $\delta^{15}N = \epsilon \ln f + \delta^{15}N_{initial}$. Herein we use an $\epsilon$ value of +15 ‰ as an average value for denitrification (Granger & Wankel, 2016).

**2.3   Pore gas nitrous oxide concentrations**

Samples of pore gas from 2 m bsd were collected every 2 weeks or every 2 months from April to November, 2013 (except when groundwater rise saturated the lower depth intervals). Samples were drawn from the subsurface using a peristaltic pump (flow rate 2 $cm^3$ $s^{-1}$). Following purging of at least 3 volumes of the sampling apparatus, the effluent end of the tubing from the peristaltic pump was attached to a 60 ml syringe and allowed to fill. The resulting gas sample was then injected





into a pre-evacuated serum vials sealed with 14 mm-thick chlorobutyl septa (Bellco Glass, Inc.) that were then shipped to Lawrence Berkeley National Laboratory for analyses. Concentrations of $N_2O$ in the samples were analyzed using a Shimadzu Gas Chromatograph (GC-2014). 4.5 ml of gas from the sample bottles was flushed through a 1 ml stainless steel loop and injected into the GC where the gases were separated on a HayeSep-D packed column (4 m x 1/8) and analyzed using an electron capture detector. The detection limit is 0.2 ppmv and the precision of the measurements approximately $\pm 10$ % of the measured value.

## 2.4 Coupled nitrifier-denitrifier model

We developed and applied a microbial trait-based model to evaluate the contributions of nitrifiers (ammonia-oxidizing bacteria and archaea and nitrite oxidizing bacteria), denitrifers, and anammox to $N_2$ cycling at the capillary fringe. The model framework is based on previously published microbial models (Bouskill et al., 2012, Le Roux et al., 2016), and represents an ecosystem of interacting functional microbial guilds competing for different carbon and nitrogen sources (Fig. 1). The different microbial groups include facultative aerobes (including denitrifiers), obligate chemolithoautotrophs (aerobic and anaerobic ammonia-oxidizing organisms) and mixotrophs (nitrite-oxidizing bacteria) (Fig. S1). The equations describing the model structure are given in the supplemental materials. Below we describe the main features of the model, the model spin-up, and the experimental simulations.

We consider three hypothetical electron donor pools (denoted $ED_{1,2,3}$) that differ in their free energy and C:N stoichiometry (C:N range = 5 - 15), in addition to separate pools of $NH_4^+$, $NO_2^-$, $NO_3^-$. In the current model, mineralization of N from OM pools is the sole source of $NH_4^+$. Gaseous pools of carbon dioxide ($CO_2$), $N_2O$, $O_2$ are also explicitly represented. The model is spatially discretized, allowing for the simultaneous representation of the three depths of the Rifle vadose zone from which the measurements were made (i.e., 2, 2.5 and 3 m). At each depth $O_2$ concentration can be manipulated by adjusting the air-filled pore space within the soil; as the water table rises, the amount of air-filled pore space declines. Diffusion of $O_2$ between gaseous and liquid phases follows Fick's law.

We represent the diversity within the different microbial groups by grouping functionally similar individuals into guilds (defined here as discrete collection of organisms performing common metabolic processes). In this respect the model represents ecological strategies that attempt to encompass variance in trait space rather than specific phylogenies. The four guilds within the facultative aerobic group represent, (1) organisms with complete denitrification pathways ($NO_3^- \rightarrow N_2$), and high $O_2$ affinity, (2) facultative aerobes with low $O_2$ affinity (i.e., switching to denitrification pathways at higher $O_2$ concentrations), (3) partial denitrifiers mediating $NO_3^- \rightarrow N_2O$, but not $N_2$, with the trade-off of a marginally higher growth rate, and (4) $N_2O$ reducers ($N_2O \rightarrow N_2$, (Jones et al., 2012; Sanford et al., 2012). Within each of the four heterotrophic guilds are three ecotypes that couple electron acceptors (e.g., $O_2$ or $NO_3^-$) to one of three electron donors ($ED_{1,2,3}$). The current iteration of the model represents $ED_{1,2,3}$ as dissolved species and does not represent extracellular enzyme production or the breakdown of more complex organic molecules. Ecotypes fall across a generalist to specialist spectrum, where organisms either specialize on one electron donor (in the present case, $ED_1$) or are able to utilize two or all three donors. The trade-off for the metabolic flexibility of a generalist is slower and less efficient growth, potentially restricting the competitiveness of this ecotype when





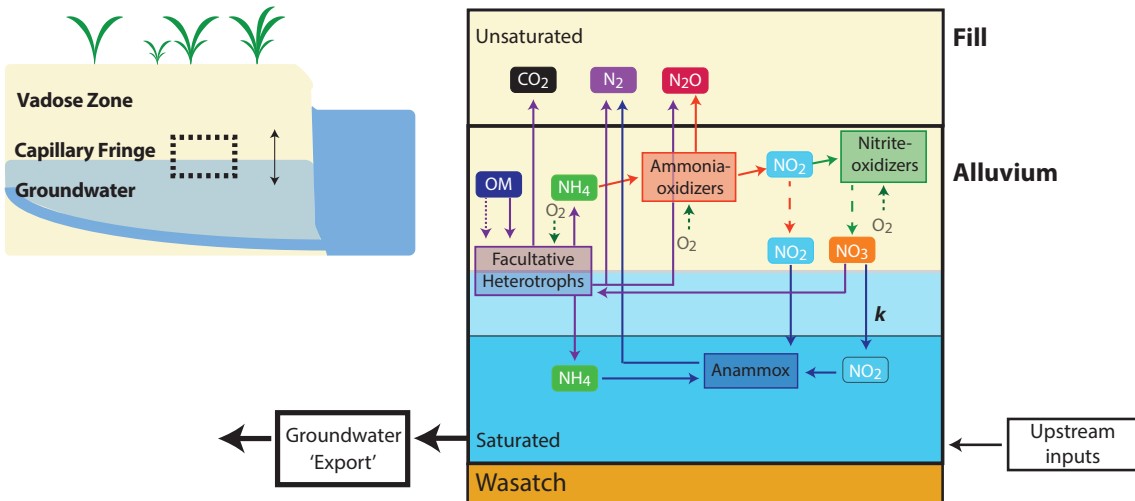

**Figure 1.** Trait-based microbial model structure showing the metabolic interdependencies represented by the model. $k$ represents the first order reduction of nitrate to nitrite used as an electron acceptor by anammox bacteria. The community is initialized within the the unsaturated zone of the Rifle aquifer prior to the rise of the groundwater (denoted by a dotted square).

only $ED_1$ is the available OM source. For heterotrophic organisms, energy for growth and biomass development is calculated based on the ATP equivalents generated by combining a given electron donor with a specific electron acceptor (e.g., $O_2$, $NO_3^-$, $N_2O$). The thermodynamic balance is calculated using a previously published approach (LeRowe et al., 2012) and presented in the supplemental material (table S3).

5     The autotrophs (AOO, NOB and anammox) possess simpler metabolisms, generating energy for the fixation of $CO_2$ to biomass via the aerobic oxidation of $NH_4^+$ (AOO, subdivided between distinct metabolisms representing the AOB and AOA), or $NO_2^-$ (NOB) or through the anaerobic oxidation of $NH_4^+$ with $NO_2^-$ (anammox). In addition to this metabolic distinction, we represent the autotrophic organisms as having both a growth rate and biomass yield (i.e., the number of moles of substrate required to build a mole of biomass) significantly lower than that of the heterotrophs (Table S1). The current code does not

10     explicitly represent $NO_3^-$-dependent metabolisms (e.g., sulfide- or iron oxidation) that are prevalent within the groundwater at this site and are a likely source of the $NO_2^-$ used by anammox. In the current model, this reaction is generally represented as a first order constant ($\sim 50\ \mu\text{M h}^{-1}$, Trimmer et al., 2005; Thamdrup et al., 2002) reducing $NO_3^-$ to a $NO_2^-$ pool that is also a function of the rate of nitrate reduction by facultative heterotrophs. Finally, mortality for all guilds is represented as a first order decay rate (in the range of $1\text{x}10^{-8}$ and $1\text{x}10^{-10}$ M d$^{-1}$).

15     **Model spin-up**: The model was spun-up for 6 months prior to the period of study (i.e., through winter conditions) in order to initialize biomasses and determine starting community composition. The spin-up simulations were run under aerobic conditions with bi-weekly inputs of $ED_{1-3}$ (to a total concentration of $\sim 1\text{x}10^{-5}$M) to foster coupling between OM mineralization and



nitrification and drive the accumulation of nitrate in the sediment by spring time (between 4.5 and 6 mM). Following spin-up, and the accumulation of $NO_3^-$ to mM concentrations, the communities were perturbed by simulating a water table rise. Under these conditions the rapid consumption of $O_2$ can rapidly lead to anoxia due to slow diffusion between the gaseous and liquid phases, and the stimulation of anaerobic metabolisms. The model outputs simulated trends in microbial functional group

composition and contribution to biogeochemical processes of interest, including the production of $N_2O$ and $N_2$.

**Model simulations**: Simulations focus on the biogeochemical and community compositional responses to rising groundwater across the capillary fringe. Therefore, these simulations do not try to represent the effect of groundwater dilution on $NO_3^-$ concentrations, however, the saturating groundwater was $NO_3^-$-deplete. For these simulations the code employed a defined grid of 0.5x0.5 $cm^2$ for each of the three depths. Starting nutrient concentrations and microbial biomass for the different depths

(2, 2.5 and 3 m) were determined by the depth-specific spin-up. For the simulations we prescribed concentrations of electron donor based on measurements of total organic matter at depth (Tokunaga et al., 2016), assuming 10 % of the total was available for microbial mineralization. Organic matter concentrations decreased with depth from 2 to 3 m, consistent with the structure of spatially discrete hotspots identified at the Rifle site (Wainwright et al., 2016). Total electron donor concentration at 2 m bsd was of $\sim 1\times10^{-4}$ M split between $ED_{1,2,3}$ in a 50:25:25 ratio, declining to $\sim 1\times10^{-5}$ M at 3 m with the same ratio. In addition,

because organic matter concentration being a proximate control on the rates of denitrification, total concentrations and ratios of $ED_{1,2,3}$ were further varied to examine the impact on denitrification.

$O_2$ concentrations within the grid cells are dependent on consumption rates, diffusion between the phases, and the replenishment rate (from top-down atmospheric diffusion), which differs dependent on the degree of saturation of the grid cell. In addition, groundwater rising through the capillary fringe entrains oxygen trapped in the pore-space and therefore have elevated

DO levels (up to 2 mg/L) relative to the groundwater below the capillary fringe (Yabusaki et al., 2017), which can inhibit denitrification. For the current simulations, a sensitivity analysis was used to determine the $O_2$ concentration of the rising water table at the three different depths. Through these preliminary studies we established the following $O_2$ conditions: At 2.5 and 3 m, the rising groundwater was both oxic ($\sim 2$ mg $L^{-1}$). However, at the shallowest depth (2 m), initial groundwater $O_2$ was slightly higher ($\sim 2.5$ mg $L^{-1}$) to also reflect the likely diffusion of atmospheric $O_2$. Each simulation was replicated 10 times

and performed at a constant temperature reflective of *in situ* conditions (13 $^o$C).

**Nitrogen loss across redox gradients**: Recent work has inferred a potentially significant role of anammox bacteria to nitrogen loss in estuarine (Song et al., 2013) and groundwater (Zhu et al., 2013) ecosystems. Previous work in the subsurface at Rifle has noted high overall abundance of chemolithoautotrophic microorganisms (Handley et al., 2012; Hug et al., 2015), and the presence of anammox planctomycetes (Castelle et al., 2013; Jewell et al., 2016). We therefore sought to clarify the

conditions under which anammox might become functionally significant across tight redoxclines found within subsurface aquifers. We simulated a series of hypothetical chemical gradients to examine the succession of denitrifier and anammox community composition. The gradients examined included monotonic increases in electron donor concentration (from $\mu$M to mM) and stoichiometry (from C:N ratios of 3 to 15, following recent work by Koeve et al., 2010 and Babbin et al., 2014), in addition to gradients in $NO_2^-$ concentrations representative of $NO_2^-$ produced through $NO_3^-$-dependent metabolisms within




the Rifle groundwater (Jewell et al., 2016). We report the trajectory of different community members and chemical species across these modeled gradients.

## 3 Results

### 3.1 Pore water chemistry

5 Over the course of the water table excursion from March to October 2014, mean values of groundwater temperature, pH, specific conductivity, oxidation reduction potential, and dissolved oxygen were 13.5 $^o$C, 7.1, 2.3 mS cm$^{-1}$, -23.1 mV, and 16 $\mu$ M, respectively. Measured NO$_3$$^-$ concentrations ranged with depth from low $\mu$M concentrations (between 2 $\mu$M and 1.8 mM) in the first 1.5 m, to 6 mM at 2.5 m (Fig. 2). NO$_3$$^-$ accumulated to 5.6, 6.1 and 4.5 mM at 2, 2.5 and 3 m respectively, and showed clear seasonal trends concomitant with changes in water table height (Fig. 2). As the water table rose (during May 10 at 2 and 2.5 m and April for 3 m), NO$_3$$^-$ concentrations declined. The onset of NO$_3$$^-$ loss was temporally offset between the different depths, occurring first at 3 m bsd (around mid to late April), then at 2.5 m (early May), and finally at 2 m by mid May. This is consistent with the timing of groundwater incursion into the unsaturated zone (Fig. 2).

NO$_2$$^-$ concentrations ranged from undetectable to 0.23 mM between 2 and 3 m bsd. While variable throughout the year NO$_2$$^-$ peaked during mid-May at both 2.5 and 3 m, at 0.23 and 0.22 mM, respectively, These peaks are temporally lagged 15 relative to the observed decline in NO$_3$$^-$ (Fig. S2). Ammonium concentrations were typically below detection limits ($\sim$ 2 $\mu$M) throughout most of the sampling period. When NH$_4$$^+$ could be measured, concentrations ranged from 5 to 85 $\mu$M.

### 3.2 Isotopic trends across the year

Prior to the onset of NO$_3$$^-$ loss, $\delta^{15}$N$_{NO_3}$ and $\delta^{18}$O$_{NO_3}$ averaged -1.8 $\pm$ 0.1 and -8.1 $\pm$ 0.3 ‰ at 2 m bsd, 3.5 $\pm$ 0.3 and -6.3 $\pm$ 1.2 ‰ at 2.5 m, and 3.8 $\pm$ 0.01 and -7.3 $\pm$ 0.1 ‰ at 3 m, respectively (Fig. 3a). A simple model (see supplemental material), 20 parameterized using calculated and literature values of the $\delta^{18}$O of different end members (nitrification and rainfall), estimates between 82.6 and 99 % of the NO$_3$$^-$ accumulating in the sediment was attributable to nitrification.

At 2 m bsd, as NO$_3$$^-$ declined there was a corresponding enrichment in $\delta^{15}$N$_{NO_3}$ and $\delta^{18}$O$_{NO_3}$ indicative of NO$_3$$^-$ turnover by an actively fractionating process (e.g., nitrate reduction). This process enriched $\delta^{15}$N$_{NO_3}$ from initial values (1.8 $\pm$ 0.1) to 7.9 $\pm$ 0.3 ‰ and $\delta^{18}$O$_{NO_3}$ from -8.1 $\pm$ 0.3 to -3.5 ‰ as NO$_3$$^-$ declined (Fig. 3a). As NO$_3$$^-$ begins to accumulate again, 25 $\delta^{15}$N$_{NO_3}$ and $\delta^{18}$O$_{NO_3}$ dropped to $\sim$ 2.3 $\pm$ 0.21 and -5.1 $\pm$ 0.27 ‰, respectively (Fig. 3a).

By contrast, at 3 m bgs a decline in NO$_3$$^-$ concentrations invoked no immediate change in the $\delta^{15}$N$_{NO_3}$ and $\delta^{18}$O$_{NO_3}$, which is indicative of dilution as NO$_3$$^-$-depleted groundwater mixes with NO$_3$$^-$ enriched porewater (Fig. 3a). A further drop in NO$_3$$^-$ after May 23rd was accompanied by an increase in $\delta^{15}$N$_{NO_3}$ from 3.5 $\pm$ 0.3 to 11.1 $\pm$ 0.3 ‰ and $\delta^{18}$O$_{NO_3}$ from -6.3 $\pm$ 1.2 to -1.8 $\pm$ 0.4 ‰, which could be the result of nitrate reduction as this depth becomes anoxic. The 2.5 m depth shows 30 further evidence of dilution/ nitrate reduction being responsible for the decline in NO$_3$$^-$. At this depth NO$_3$$^-$ concentrations declines with little initial impact on $\delta^{15}$N$_{NO_3}$ and $\delta^{18}$O$_{NO_3}$. However, after June 6th isotope enrichment was observed in the





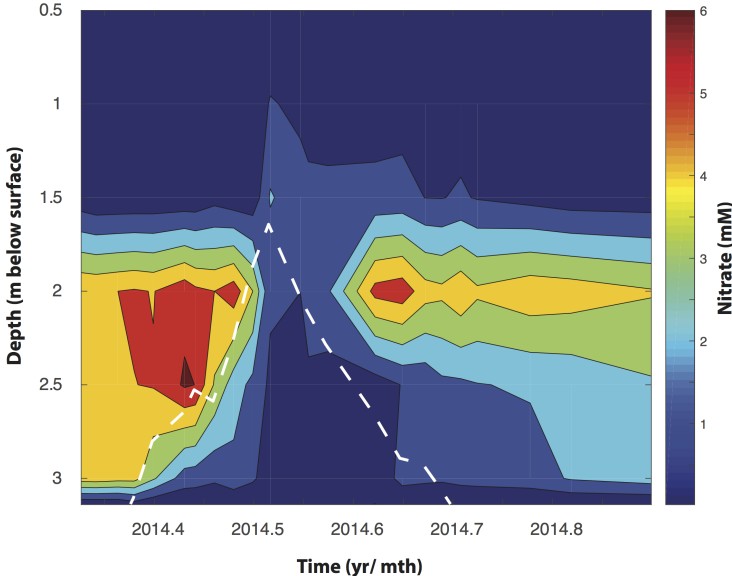

**Figure 2.** Depth resolved $NO_3^-$ concentrations changes in the Rifle floodplain well (TT-03) during an annual rise watertable depth (shown by the dotted white line).

residual $NO_3^-$ increasing from $\sim 4.0 \pm 0.1$ to $8.4 \pm 0.2\,‰$ and, from $-7.2 \pm 0.1$ to $-2.8 \pm 0.1\,‰$ for $\delta^{15}N_{NO_3}$ and $\delta^{18}O_{NO_3}$, respectively.

A simple Rayleigh model was used to estimate the contribution of actively fractionating processes to the observed drop in $NO_3^-$ concentrations at each depth. At 2 m, a shift in $\delta^{15}N_{NO_3}$ of $+10\,‰$ suggests nitrate reduction is responsible for $\sim 64$ % of the drop in $NO_3^-$ concentrations. By contrast, dilution appeared to dominate at 2.5 and nitrate reduction was calculated to be responsible for only $\sim 28$ % of the observed decline in $NO_3^-$ concentrations. At 3 m bsd the initial decline in $NO_3^-$ is followed by a second period of $NO_3^-$ draw down. Approximately 91 % of the $NO_3^-$ loss during this first event was attributable to dilution. Conversely, the contribution of dilution declines to 53 % of the second event with the rest attributed to nitrate reduction.

## 3.3 Pore gas N₂O concentrations

The concentrations of $N_2O$ measured in the unsaturated zone above the water table were elevated relative to atmosphere throughout the study period. At the 2 m depth, $N_2O$ peaked at 25 ppm between late May and early June, concomitant with the rise of the water table during the most intense periods of $NO_3^-$ decline (Fig. 3b). The $N_2O$ concentration at the 2.5 m depth peaked at 35 ppm in late July shortly after the water table dropped below this depth and when the $\delta^{15}N$ of the nitrate in the pore water at this depth reached its highest (most reduced) value. The 3 m interval was saturated during the period from late-April through late-October and no $N_2O$ samples were taken during this period of intense denitrification at this depth.





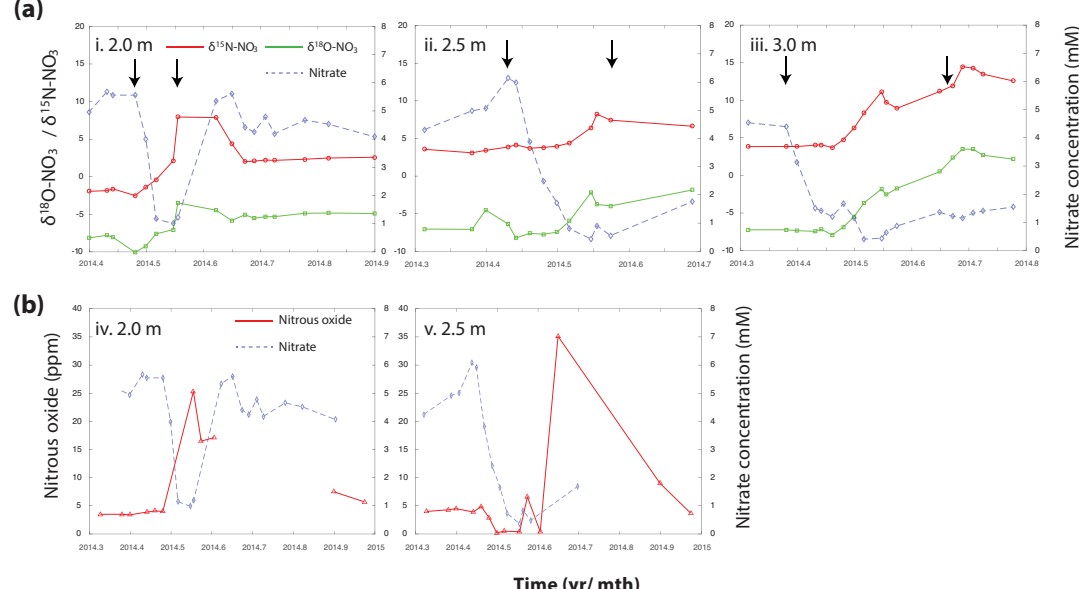

**Figure 3.** (a) Changes in $NO_3^-$ concentrations, and the associated $\delta^{15}N_{NO_3}$ and $\delta^{18}O_{NO_3}$ over the Spring and Summer of 2014 at (i) 2 m, (ii) 2.5 m and (iii) 3 m bsd. (b) Trajectory of $N_2O$ trace gas measurements against $NO_3^-$ concentrations across the same depth profile, (iv) 2 m, and (v) 2.5 m. These measurements were not taken as frequently than the $NO_3^-$ isotopic measurements but encompass the period depicted in the corresponding panels above. Arrows indicate the period during which the water table saturated the relevant depth.

### 3.4 Simulated microbial community response to water table rise

Initial model simulations sought to replicate the $NO_3^-$ trends during groundwater fluctuations and shed light on the interactions and roles of the microbial community members. Model simulations were initialized following the aforementioned spin-up period and under the assumption that the pore-space was saturated with groundwater. The simulations proceed over a period of variable saturation as the groundwater begins to fall. The period of saturation is specific to each depth, with the shallowest depth (2 m bsd) having the lowest saturation period. $O_2$ in the saturated pore space is consumed rapidly over a short period of time due to consumption by facultative aerobes and nitrifiers, and little replenishment via diffusion.

Under aquifer-relevant organic matter concentrations at 2 m bsd, aerobic consumption occurs rapidly with the onset of denitrification almost immediate. The pore space remains anoxic for the duration of saturation, and under these conditions and initial organic matter concentrations, denitrification accounts for 50 % of the initial $NO_3^-$ (Fig. 4a), consistent with the proportion calculated from observations using a Rayleigh model.

This $NO_3^-$ was consumed mainly by the heterotrophic denitrifiers, at a rate similar to observations. Within this community metabolic specialists (solely consuming $ED_1$) mediating both partial (to $N_2O$) and complete (to $N_2$) denitrification were most responsive to changes in $O_2$ concentrations, indicating that the aforementioned trade-off, whereby partial denitrifiers ($NO_3^-$ $\rightarrow N_2O$) have a slightly higher growth rate, does not necessarily offer an advantage (Fig. 4b). The relative abundance of





anammox organisms emerging in the model under anoxic conditions was always a minor proportion of the overall community (from $5\%$ at the simulation initialization to $\sim 3.4\%$ by the end of the simulation). However, anammox planctomycetes mainly grew during the most intense periods of denitrification, likely consuming the $NH_4^+$ released from OM mineralization under anaerobic conditions (see Fig. S3 and a extended discussion below). The anammox planctomycetes were therefore responsible

for nitrogen loss from the aquifer, facilitated by heterotrophic activity. As the groundwater receded, and the degree of pore space saturated fell, $O_2$ diffused back into the pore stimulating nitrification consuming $NH_4^+$ culminating in the subsequent accumulation of $NO_3^-$ (Fig. S3). The model captures the qualitative trend here, but fails to capture the rate at which $NO_3^-$ accumulates. Increasing the abundance of nitrifiers (by manually increasing the initialized population concentration to 20 % of the community) improved the consistency between observations and modeled $NO_3^-$ response (Fig S4). However, this approach

is unlikely to represent the *in situ* abundance of nitrifiers (Castelle et al., 2013; Hug et a., 2015).

At 2.5 m, the higher dissolved $O_2$ concentrations delay the onset of denitrification relative to the 2 m depth, however, anoxia appears to be reached more rapidly that observations suggest. Furthermore, $NO_3^-$ turnover via denitrifying heterotrophs accounts for approximately 34 % of $NO_3^-$ loss, which is similar to that estimated from the isotopic data, with the slight overestimation likely to higher modeled organic matter concentrations. Heterotrophic bacteria again dominated community composi-

tion at this depth, with specialist metabolisms carrying out partial and complete denitrification generally dominating. However, over time the substrate for the specialists ($ED_1$) decreases in concentration, facilitating an increase in both intermediate and generalist metabolisms. At this depth, anammox bacteria were initially at low abundance, but increased to $\sim 7\%$ to total population abundance after a brief increase in $O_2$. It is likely this larger anammox population is due to the longer period of anoxia at 2.5 m compared with the 2 m depth facilitating an increasing abundance of characteristically slow growing organisms as

$NH_4^+$ is released during heterotrophic mineralization.

As with the two shallower depths, denitrifiers dominated nitrogen loss at 3 m, with a sharp increase in the heterotrophic specialists (i.e., organisms than consume only $ED_1$) at what appears to be the onset of denitrification. However, denitrification was also overestimated at this depth relative to Rayleigh calculated $NO_3^-$ reduction. Model simulations within aquifer-relevant organic matter concentrations suggest 30 % of $NO_3^-$-reduciton was attributable to biological processes.

The model simulations demonstrate the importance of organic matter concentration as a proximate control on the rate and extent of denitrification, and indicate that carbon limitation, in addition to the length of the anoxic period, can explain denitrification accounting for only a third to two-thirds of the measured decline in $NO_3^-$ concentrations. For example, the trends in measured $NO_3^-$ concentrations can be matched simply by increasing the initial electron donor concentration an order of magnitude (Fig. S4). This results in 100% of $NO_3^-$ loss being attributable to biological nitrogen turnover, contrary

to the observations.Furthermore, the relative ratio of electron donors can also influence the extent of denitrification. Ratio that favor larger concentrations of $ED_3$ reduces the extent of denitrification during the anoxic period toward that observed above (Fig. S5). This emphasizes the need for the further establishment of qualitative and quantitative benchmarks to test trait-based models. In this case, isotopic fractionation appears to be a promising benchmark for such models.

Predictions of $N_2O$ production in the model are qualitatively consistent with observations at 2 and 2.5 m bgs, and increase

during the onset of denitrification (Fig. S3). However, quantitative differences emerge. At 2 and 2.5 m predicted concentrations



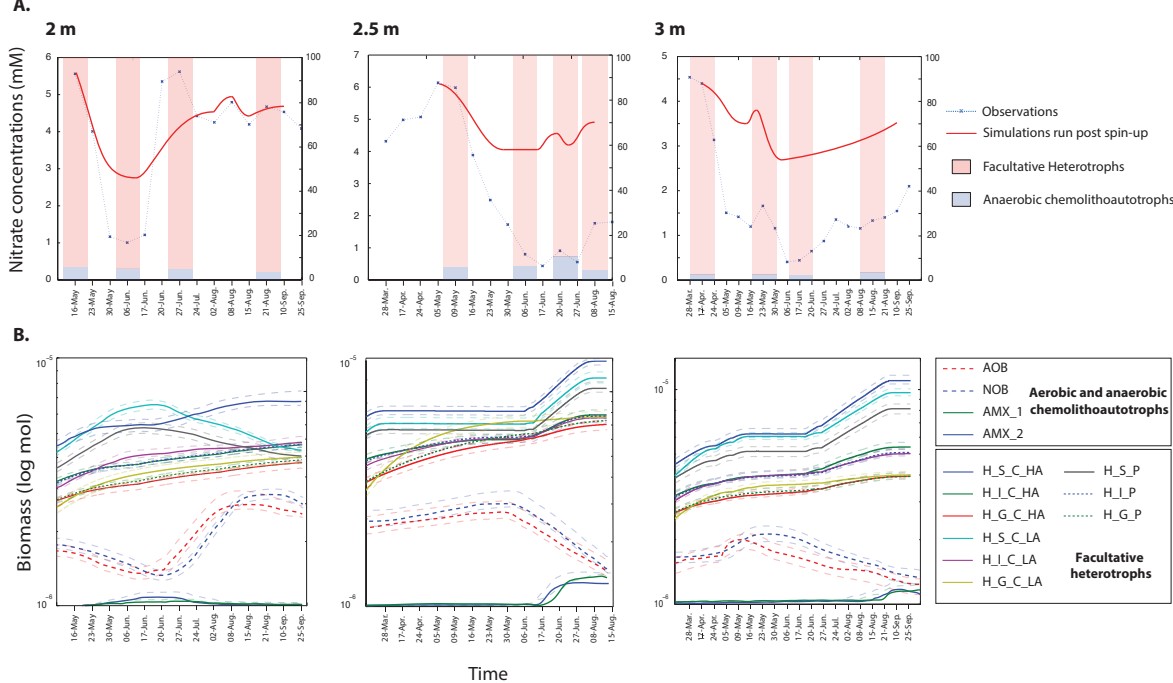

**Figure 4.** Microbial model simulations of nitrate turnover and associated trends in community emergence and community trajectories. The model was run under aquifer-relevant organic matter concentrations to reproduce trends in the measured $NO_3^-$ data at 2 m, 2.5 m and 3 m bsd. The top panel (a) shows the measured and simulated $NO_3^-$ concentrations and overall community composition split between heterotrophs and chemolithoautotrophs. The simulations solely represent the biological reduction of $NO_3^-$, and do not include dilution. The bottom panel (b) breaks down the active biomass representative of the overall community composition into distinct autotrophic and heterotrophic functional guilds. Abbreviations: AOB: Ammonia oxidizing bacteria (sum of four guilds). NOB: nitrite-oxidizing bacteria (sum of three guilds). AMX: Anammox. The heterotrophic organisms are labeled depending on metabolism. H = Heterotroph. S = Specialist. I = intermediate. G = Generalist. C = Complete denitrification (to $N_2$). P = Partial denitrification (to $N_2O$). LA = low $O_2$ affinity. HA = high $O_2$ affinity.



of $N_2O$ are higher, at 110 and 61 ppm respectively, than measured concentrations (Fig. 3a/b). At 3 m depth, $N_2O$ production also shows a peak corresponding with denitrification that is contrary to concentration measurements from this depth.

### 3.5 Nitrogen loss across redox gradients

Under conditions of low OM C:N stoichiometry ($\sim$ 3), and with no consideration for allochthonous inputs of substrate (e.g.,
$NH_4^+$), anammox contributes a maximum of 30.6 % to the total nitrogen loss (i.e., the sum of $N_2O$ and $N_2$) across a range in
prescribed $NO_2^-$ concentrations (from $\mu$M to mM concentrations, Fig. 5). The contribution of anammox falls to zero as C:N
stoichiometry increases and the prescribed $NO_2^-$ concentrations decline. Across the OM/ $NO_2^-$ gradient, the contribution
of anammox to nitrogen loss shows a bimodal distribution. Anammox contributes significantly to nitrogen loss (> 20 %)
under conditions where denitrification is limited by electron donor concentration, or where anammox is closely coupled to
denitrification. The first scenario leads to very low rates of N loss dependent on the initialized $NH_4^+$ concentrations. The
latter scenario can sustain coupled $N_2$ loss as long as OM inputs are maintained, facilitating the development of a commensal
relationship between the two different guilds. This relationship weakens as OM concentrations increase (in this case above a
value of $\sim$ 10 $\mu$M OM concentration) allowing denitrification to dominate nitrogen loss, or as the OM stoichiometry increases
and the relative concentration of $NH_4^+$ released during mineralization declines (Fig. S6). Under these conditions, the relative
importance of anammox as a pathway for nitrogen loss declines, however, anammox biomass does not decline (data not shown),
and in many cases increases as more $NH_4^+$ becomes available.

## 4 Discussion

### 4.1 Coupled nitrification and denitrification drive subsurface nitrogen cycling

Sharp redoxclines created at terrestrial-aquatic interfaces are potential hotspots of nitrogen loss (Lohse et al., 2009). Previous
observations have noted the annual accumulation and dissipation of $NO_3^-$ at the capillary fringe of the Rifle site. Measurements
presented here imply a combination of both physical (i.e., $NO_3^-$ dilution by rising groundwater) and biological processes
(autotrophic nitrification and heterotrophic denitrification) are responsible for the nitrogen dynamics observed at the capillary
fringe and are driven by the changing water table levels.

#### 4.1.1 Nitrate accumulation in the vadose zone:

At the outset of the study, just prior to snow melt, $NO_3^-$ concentrations ranged 4 to 5.5 mM, and $\delta^{15}N_{NO_3}$ and $\delta^{18}O_{NO_3}$
measurements indicate that coupled ammonia and nitrite oxidation (nitrification) were responsible for the majority of the $NO_3^-$
accumulated under oxygenated conditions in the vadose zone. A simple two end-member mixing model (see supplemental
material) was parameterized using calculated and literature-derived values for $\delta^{18}O_{NO_3}$ of different end members. These
included the $\delta^{18}O$ of $NO_3^-$ deposited via rain/ snowfall over Colorado ($\sim$ 68.3 ‰ Kendall et al., 2007), and a range of $\delta^{18}O$
values for nitrification (calculated using $\delta^{18}O$ of site water and a previously described method, Fang et al., 2012). Using this




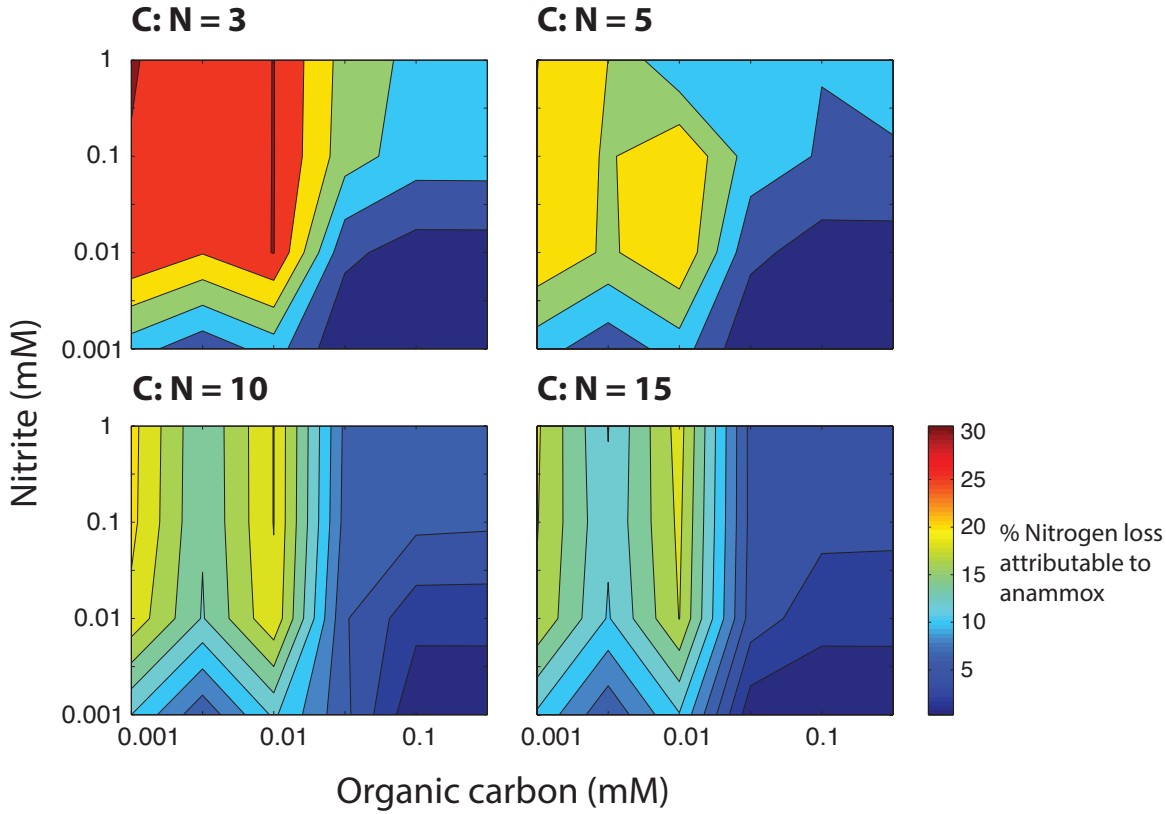

**Figure 5.** Metabolisms involved in nitrogen loss across gradients in OM and $NO_2$ concentrations and across OM compounds with varying C:N stoichiometry. The data is presented as the percentage contribution of chemolithoautotrophic anammox to nitrogen loss relative to denitrification.

model we estimate that nitrification contributed between 82.5 and 97.5 % of the $NO_3^-$ within the subsurface. This value is plausible given the low rates of net recharge from surface precipitation ($\sim$ 3 cm yr$^{-1}$, Yabusaki et a., 2017).

Metagenomic surveys of the Rifle floodplain indicate that ammonia-oxidation is likely carried out by the Thaumarchaeota (AOA) (Castelle et al., 2013; Hug et a., 2015). This is unsurprising given the low $NH_4^+$ concentrations in the Rifle subsurface,
5   as the AOA have previously been characterized as having half-saturation constants in the nM range (Martens-Habbena et al., 2009) and dominate nitrification in ecosystems where $NH_4^+$ concentrations are similarly low (Beman et al., 2012). The source of $NH_4^+$ supporting nitrification at the capillary fringe has yet to be identified, and while sediment adsorbed $NH_4^+$ (Böhlke et al., 2006) and biological nitrogen fixation (Swanner & Templeton, 2011; Lau et al., 2014) might be a source of subsurface nitrogen it is more likely that OM mineralization contributes the bulk of $NH_4^+$. Rifle sediments can be carbon rich due to the
10   advective downward transport of DOM (Tokunaga et al., 2016), and heterogeneously distributed fine-grained sediment lenses (Janot et al., 2016). These lenses are enriched in organic carbon that likely represent the deep burial of soil horizons (Janot et al., 2016), common within floodplain sediments (Blazejewski et al., 2009; Hill, 2010).





### 4.1.2 Nitrate loss during groundwater rise:

During the spring snow melt the Colorado River rises, increasing the height of the groundwater table in the adjacent Rifle floodplain. The temporal offset in $NO_3^-$ decline between 3, 2.5 and 2 m bsd (Fig. 2) indicates that the groundwater rise and resultant change in $O_2$ availability are primarily responsible hot moments of biogeochemical activity (McClain et al., 2003). This appears to be the case 2 m bsd, a depth not annually saturated by rising groundwater. Here, a steady increase in the $\delta^{15}N_{NO_3}$ and $\delta^{18}O_{NO_3}$ accompany the rapid fall in $NO_3^-$ concentrations, and is indicative of nitrate reduction (further discussed below). At both 2.5 and 3 m, the onset of $NO_3^-$ loss precedes the enrichment of $\delta^{15}N_{NO_3}$ and $\delta^{18}O_{NO_3}$, indicative of dilution of $NO_3^-$ as the groundwater rises. While the groundwater at Rifle is anoxic, as it rises into the unsaturated zone it sequentially entrains $O_2$ at the interface of the groundwater and the unsaturated zone (Yabusaki et al., 2017). This subsequent oxygenation can inhibit denitrification, which likely delays the onset of denitrification at the deeper depths. However, the 2 m depth does not follow this trajectory, despite the likelihood of $O_2$ entrainment at this depth. It is plausible that a higher rate of activity at this depth is responsible for the rapid consumption of $O_2$ upon saturation stimulating the onset of nitrate reduction, however, further work is required to verify this. The metabolisms responsible for the turnover of $NO_3^-$ at the capillary fringe are difficult to distinguish through conventional molecular methods. Several recent studies have highlighted the predominance of chemolithoautotrophic metabolisms within the groundwater, including dissimilatory nitrate reduction to ammonia (DNRA), $NO_3^-$-dependent iron oxidation, via the Gallionellaceae, and anammox (Hug et al., 2015; Jewell et al., 2016). A question therefore remains as to whether these metabolisms translocate as the water table rises, and contribute to $NO_3^-$ and $NO_2^-$ turnover, or whether nitrate is cycled at the capillary fringe by heterotrophic denitrifiers. At the capillary fringe it appears more likely that heterotrophic denitrifiers catalyze the bulk of biological $NO_3^-$ turnover, with a smaller role for chemolithoautotrophic metabolisms (i.e., $NO_3^-$ reduction coupled to anammox). There are several reasons to speculate in this way. Firstly, recent studies of the molecular microbial diversity of the Rifle subsurface demonstrate a broad distribution of heterotrophs capable of carrying out nitrate reduction and additional denitrification pathways from the vadose zone into the groundwater (Hug et al., 2015; Anantharaman et al., 2016), but a general restriction of anaerobic chemolithoautotrophs to the groundwater and naturally reduced zones within the floodplain (Jewell et al., 2016).

Secondly, the measured in $N_2O$ peak 2 m below the surface coincides with the most intense period of $NO_3^-$ and $NO_2^-$ removal (Fig. 3 and S2). $N_2O$ is produced as an intermediate during denitrification or nitrifier denitrification. To our knowledge, $N_2O$ is not an intermediate produced during anaerobic ammonia oxidation. The conditions that partition $N_2O$ flux between heterotrophic denitrification and autotrophic nitrite denitrification are not well defined, however, concurrent measurements of the $\delta^{15}N_{N2O}$ and $\delta^{18}O_{N2O}$ indicate heterotrophic denitrification is the major source of $N_2O$ in the unsaturated zone (Bill et al., in prep).

To further parse out the potential contributions of different metabolisms to nitrogen turnover we developed a microbial model based on the functional and physiological traits of the different guilds at the capillary fringe. Under conditions replicating the Rifle capillary fringe and using site-specific data to initialize the physical environment, the competitive dominance of heterotrophic denitrifiers is clear. While this dominance is shaped somewhat by the initial conditions established by an aerobic





spin-up, which would select against anammox, it is also attributable to the very slow growth rate of anammox bacteria, consistent with ecophysiological understanding (Kartal et al., 2011). Furthermore, when initializing with a higher chemolithoautotrophic population, mimicking the translocation of groundwater populations with the water table rise, anammox is still rapidly outcompeted for $NH_4^+$ and $NO_2^-$ by the heterotrophic organisms under conditions found in the well.

At 2 m, heterotrophic specialists (characterized by the ability to grow rapidly on one electron donor) respond faster than heterotrophic generalists (guilds that can use 2 or more electron donors, albeit at a lower growth rate) to changes in oxygen concentrations by switching from aerobic to anaerobic metabolisms and consuming $NO_3^-$. More specifically, partial denitrifiers ($NO_3^- \rightarrow N_2O$), able to grow a little quicker, outcompete other metabolic specialists. This is somewhat consistent with reports of specialists dominating microbial communities across different habitats (Mariadassou et al., 2015). However, as

the simulation progresses and the relevant electron donor is depleted, metabolically flexible organisms become more prominent within the community. At 2.5 and 3 m, the geochemistry changes more slowly than 2 m, due to the potential for DO appropriated from the pore space to support aerobic activity for longer. Under these conditions, the onset of anaerobic conditions gradually emerges relative to 2 m, however, the heterotrophic populations generally respond in step with one another, preserving a similar community composition.

Conversely, while the transcriptional up-regulation of anammox bacteria can be rapid (Jewell et al., 2016), the anammox planctomycetes are characterized by slow growth rates ($\sim$ 0.0026 - 0.0041 $h^{-1}$) and a thermodynamically limiting metabolism (Kartal et al., 2011; 2012), impinging on the rate at which these organisms respond to changing environmental conditions. Reported rates of nitrogen loss from anammox for estuarine sediments are significantly lower than for denitrification (Trimmer et al., 2011), and are lower than rates inferred from this study. Finally, this emergent response in the microbial model is

consistent with recent observations showing a predominance of likely heterotrophic organisms (including $\beta$-proteobacteria and potential microaerophilics, Hug et al., 2015) around the capillary fringe, giving way to chemoautotrophy or symbiotic-metabolisms deeper within the aquifer (Hug et al., 2015; Yabusaki et al., 2017).

## 4.2   Simulated denitrification and anammox contributions to nitrogen loss in the subsurface

The dominance of nitrogen loss by heterotrophic denitrification in the Rifle subsurface is at odds with recent studies attributing

a significant role in the nitrogen cycle at terrestrial-aquatic interfaces to anammox bacteria (Zhu et al., 2013; Smith et al., 2015). While denitrification is largely regulated by organic matter availability it is likely that anammox activity is also closely coupled to organic matter remineralization as the source of $NH_4^+$ (Koeve and Kähler, 2010). Recent observations support the assumption that organic matter quality and quantity control the ratio of nitrogen loss via denitrification: anammox activity (Babbin et al., 2014), restricting anammox to between 24 to 43 % of nitrogen loss, similar to a previously calculated value

(Koeve and Kähler, 2010). Anammox activity can be further regulated by $NO_3^-$-reducing metabolisms controlling $NO_2^-$ supply (Trimmer et al., 2005).

    We used the microbial model to examine the contribution to nitrogen loss from denitrifiers and anammox bacteria across gradients of OM concentration and stoichiometry and $NO_2^-$ concentrations. A two-phase relationship between nitrogen loss via denitrification and anammox emerged from the initial conditions, wherein, anammox accounted for $\sim$ 15 % or more of




nitrogen loss under conditions of low OM concentrations where denitrification was minimal, or under conditions where OM mineralization provides significant $NH_4^+$ release without denitrification overwhelming the anammox signal for nitrogen loss. Under these circumstances the two processes appear coupled. However, this also necessitates a consistent source of $NO_2^-$. This is plausible within the Rifle site, where $NO_2^-$ can be supplied from the activity of nitrate-dependent iron and sulfide

oxidizers (Jewell et al., 2016). Across these gradients the specific rate of anammox does not change markedly, but their contribution to total nitrogen loss declines as denitrifier activity increases with OM concentration. At low OM concentrations, and across gradients of $NO_2^-$ input, anammox growth is limited by $NH_4^+$ availability (Fig. S6). Under higher OM concentrations, mineralization rates outweigh anammox consumption due to $NO_2^-$ limitation and $NH_4^+$ accumulates.

    These modeled insights inform our conceptual model for nitrogen cycling within the Rifle subsurface (Fig. S7). Around the

capillary fringe high organic matter concentrations (either within or proximate to the NRZs, Janot et al., 2016) support coupled nitrification and denitrification, oxidizing reduced nitrogen, released during organic matter mineralization, to $NO_3^-$, which can be reduced to $N_2O$ or $N_2$. Under the capillary fringe, high iron and sulfide concentrations within the NRZs and groundwater support chemolithoautotrophic $NO_3^-$-reduction to $NO_2^-$ that can be coupled to anammox activity (Jewell et al., 2016). Herein, DNRA could be a potential source of $NH_4^+$ below the capillary fringe (Hug et al., 2015), allowing for the decoupling of these

processes from heterotrophic mineralization. Overall we find little to suggest that these chemolithoautotrophic metabolisms transition toward the capillary fringe during a rise in the water table. We therefore conclude that while chemolithoautotrophic anammox located below the capillary fringe can be a slow and persistent sink for nitrogen (Fig. S7), denitrification occurs episodically under conditions where hot-spots form as the groundwater rises to encompass previously unsaturated sediment. This can result in the rapid reduction of large concentrations of $NO_3^-$ over short periods of time (Pinay et al., 2007; Duncan

et al., 2015).

## 5   Conclusion

Contrary to recent reports highlighting the importance of chemolithoautotrophic metabolisms (in particular aerobic ammonia and nitrite oxidation) to the nitrogen cycle at terrestrial-aquatic interfaces, we find here that nitrogen cycling around the capillary fringe within the Rifle floodplain was predominantly attributable to coupled nitrification-denitrification. Moreover, we

highlight that during the most intense periods of heterotrophic denitrification, rates of $N_2O$ produced are significantly larger than have been previously associated with subsurface aquifers (McMahon et al., 2000; Hiscock et al., 2003; Weyman et al., 2008), thus arguing for a greater consideration of such regions within global $N_2O$ budgets (Davidson and Kanter, 2014).

*Acknowledgements.* This work was performed as part of the Lawrence Berkeley National Laboratory's Sustainable Systems Scientific Focus Area funded by the U.S. Department of Energy, Office of Science, Office of Biological and Environmental Research under contract DE-

AC02-05CH11231, and in part from a Laboratory Directed Research and Development Grant from the Office of the Director at Lawrence Berkeley National Laboratory.



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
