# Peer review of "Evidence for microbial mediated nitrate cycling within floodplain sediments during groundwater fluctuations"

_Biogeosciences, 2017_

## Referee Comment (RC1) · Anonymous Referee #1 · 14 Jul 2017

The authors report a sparse dataset of mineral N and N2O concentrations and stable isotope compositions in groundwater samples collected at the Rifle site. They use these data to test a detailed mechanistic model of microbial N cycling, which predicts the nitrate concentrations relatively poorly. The main conclusion from these data appears to be that significant denitrification occurred in the sediment following groundwater rise driven by snowmelt. This finding is not novel—the importance of riparian denitrification has long been known and has received extensive study in montane, snowmelt-dominated systems. See for example the extensive work by Paul Brooks, Michelle Baker, Mark Williams, and others.

[Figure]

The nitrate isotope data are nice to see, but the marriage with the microbial model seemed forced to me. Where are the microbial data needed to test the main predictions of the model (e.g. Fig 4)? There are skilled microbial ecologists on this team and working on this site. I didn't find the model results compelling in the absence of microbial data, especially given the poor performance of the model in predicting nitrate at the three (!) depths where it was apparently compared (Fig. 4). The microbial simulations come off as entirely speculative given that there are no data presented, as does the speculation as to the importance of annamox vs. canonical denitrification vs. chemolithotrophic processes. Contrary to the conclusion (P17 25), I don't think the authors can make any concrete claims as to the mechanisms driving the patterns observed, especially given that the nitrate isotope fractions are not well constrained for these pathways, and that there is enormous variation in nitrate isotope fractionation during denitrification.

The spatial replication of the field data seems inadequate given the heterogeneity of the system under study. Why are no isotope measurements from the vadose zone and shallow soils reported? This seems critical to get at the question of biogeochemical processing of N vs. dilution or mixing that comes up throughout the paper, and the enormous spatial heterogeneity of nitrate isotope compositions that is increasingly documented in the literature. What is the composition of the water that is posited to be diluting the sediment zone of interest? There was almost no discussion of the hydrology of the site and potential source waters, which are critical for getting at this point. To interpret the isotopes, you would need to consider mixing rather than pure dilution unless you could demonstrate that you were mixing nitrate-rich vs. nitrate-free water. This is especially critical in the context of the heterogeneity in buried organic lenses that has been demonstrated at this site.

I am very surprised that the authors report nitrate concentrations of $\sim$5 mM—surely they must mean 5 micromolar or 5 mg NO3- L-1? Note that nitrate concentrations this high (5 mM) are rarely observed even in heavily contaminated agricultural streams or

sewage effluent. If this is correct, what is the source of this extremely high nitrate? It seems implausible that this would be produced via natural organic matter decomposition, and the authors report low NH4+ values. There is a rich literature on N cycling in the Rockies, and the reported values are totally out of context with this. I suspect that this is either a silly mistake or a serious technical problem.

The manuscript is riddled with errors. In the title alone there is a grammatical error and a misspelling of one of the author names. I urge the lead author to give the paper a proper proof reading before sending out for review! For example, P8 line 18, $\delta$15N is given as -1.8. At line 23, this same value is referred to but the minus sign is missing. There are many more examples...

P4 30: "Samples of pore gas from 2 m bsd" do you mean below 2 m? How many depths were sampled? P9 15: what do you mean by "highest (most reduced) value" The message in Fig. 3 is not at all clear as presented. Try putting the same values ($\delta$15N, e.g.) on a common plot so we can compare the trends among depths over time. P13 20: Need citation P15 25: "the measured in N2O peak"

---

## Referee Comment (RC2) · Anonymous Referee #2 · 15 Jul 2017

Overall Quality:

The authors attempt to create a mechanistic model based on experimental isotope data from a floodplain in Rifle, Colorado. The model for this ecosystem is unique and definitely would be helpful for biogeochemists and practitioners attempting to identify denitrifying hotspots. The dataset used in the article is within a very small time period and the model appears to be only calibrated, but not validated. Further, nitrate concentration predictions appear to not be closely correlated, but statistical evaluations for goodness of fit were not presented in the article. The main conclusion, concurred with most research in floodplain ecosystems, that denitrification was the predominant

removal mechanism for nitrate. Overall, the model in theory would be very helpful. However, the article in its present form lacks validation through microbial isotope experiments, which would truly provide "evidence" for microbial mediated nitrate cycling. Overall, the model presented in this paper has the potential to be an important step for predicting microbial processes in floodplain soils in Colorado and similar landscapes. However, validation and statistical assessments must be completed for the authors to be convincing that this model can predict all that is currently claimed.

Individual Scientific Questions/Issues:

The dataset is a small dataset and for only one season. The details of that season (e.g., rather it was a normal rainfall year) were not presented. Further, the article was unclear as to how many transects of monitoring wells or replicate samples were taken at each depth. Likely, a map of the monitoring locations would be helpful within the floodplain. Further information is needed regarding the soil and water chemistry (e.g. pH), which will impact microbial community population and productivity. The high concentrations of nitrate were surprising. Is this area in an agricultural landscape? Further information regarding the watershed land use would be helpful to understand the source of such high nitrate concentrations.

The nitrate isotope data was encouraging to see in the experimental design. However, the title "Evidence for microbial mediated nitrate cycling..." seems misleading, assuming no microbial isotopic data was collected in the soils. Additionally, do the authors have isotopic data for the confined aquifer to confirm that the mixing water is truly nitrate free?

The models need statistical results. Based on purely visual observations, the model predications do not appear to fit well with the observed data (Figure 4).

Technical Corrections:

The data presented in section 3.2 may benefit with having a table. All of the numbers

are difficult to follow and compare in their present form. The second sentence in section 4.1 needs a reference. Additionally, several of the graphs are difficult to read (e.g. Fig 4, S3, S4).

---

## Author Comment (AC1) · 12 Sep 2017

See for example the extensive work by Paul Brooks, Michelle Baker, Mark Williams

The manuscript under review examines how nitrate is produced and transformed at the capillary fringe during annual fluctuations in the water table. We are familiar with the large body of Paul Brooks' work, however, this work primarily concerns itself with the measurement and modeling of the subnivial nitrogen cycle, as well as examining surface hydrological processes. Much of this work is not necessarily pertinent to the current manuscript. Mark Williams similarly takes a broad approach to examining hydrologically-induced changes in the nitrogen cycle, however, as far as we are

aware does not work around the capillary fringe either. Michelle Baker's work primarily focuses on carbon and nitrogen cycling within riverine biogeochemical hotspots (i.e., riparian areas of rivers corridors, and in-stream hyporheic zones) rather than around the capillary fringe. While these authors' outstanding work has informed our broader thinking on the terrestrial nitrogen cycle, their body of work is not immediately applicable to the current manuscript, which is why these papers have not been cited. Indeed, there are few studies in the literature that we can find, taking a mechanistic approach to understanding the nitrogen cycle at capillary fringes that serve as relevant citations for the current work. This also goes against the reviewer's supposition our work is not novel. While there are a number of manuscripts examining nitrogen dynamics around the hyporheic zone of streams (e.g., Bohlke et al., Biogeochem. 2009; Zarnetske et al., JGR, 2011) and rivers (e.g., Clilverd et al., Biogeochem, 2008; Hinkle et al., J. Hydro. 2001), and oxygen transformations around the capillary fringe itself (Haberer et al., J. Contamin. Hydrol), we can, in fact, find few manuscripts that examines the importance of hydrological fluctuations around the capillary fringe with respect to nitrogen cycling. The manuscripts that examine nitrogen cycling around the capillary fringe (e.g., Abit et al., Geoderma, 2008), all of which are referenced in the current manuscript (Pg. 2, Ln. 5 - 6) do not take a similar mechanistic approach as described here.
* * *
Where are the microbial data needed to test the main predictions of the model (e.g. Fig 4)? There are skilled microbial ecologists on this team and working on this site. I didn't find the model results compelling in the absence of microbial data, especially given the poor performance of the model in predicting nitrate at the three (!) depths where it was apparently compared (Fig. 4).

There has been a significant amount of microbial work performed at this site (e.g., Anantharaman et al., Nat. Comm, 2016; Hug et al., ISME, 2015; Jewell et al., ISME, 2016; Wrighton et al., ISME, 2014), all of which is extremely useful for initializing the reaction network for this, and other, models (as pointed out on Pg. 3, Ln 20, and discussed further on Pg. 14). For example, one of the questions we set out to address in the current manuscript concerns the interactions between different heterotrophic and autotrophic metabolisms promoting N-transformations and loss (denitrification vs. anaerobic ammonium oxidation, see Pg. 3, Ln 5). The notion that anammox is important in this environment comes directly from molecular evidence profiling the community within the naturally reduced zones of the floodplain (e.g., Jewell et al., ISME, 2016). These areas, located just below the capillary fringe, have high abundance of chemolithoautotrophic metabolisms, however, little information exists comparing the importance of different metabolisms to nitrogen loss. Several manuscripts have tackled these questions within marine environments (e.g., Babbin et al., Science, 2014; Koeve & Kahler, Biogeosciences, 2010) using measurements and models, but as far as we are aware, this has not been extended to terrestrial systems. Furthermore, because feedbacks between biotic and abiotic systems are inherently non-linear, and therefore cannot be addressed directly by molecular studies, we believe a mathematical model of interacting microbial guilds informed by these prior studies is a plausible approach to address these interactions.

However, we have yet to find microbial data that is applicable for benchmarking microbial models. Microbial models represent the active portion of the microbial community, and are simplified representations of microbial guilds using several traits, and imposed trade-offs. Therefore, commonly collected microbial metrics are not comparable to modeled metrics. For example, measurements of biomass (via chloroformfumigation) account for microbial and fungal biomass and additional labile compounds from non-living sources (e.g., plant residue), and frequently overestimate biomass. Modeled biomass, on the other hand, represents the products of growth of the metabolisms considered (never the full community).

Molecular markers of microbial activity (e.g., mRNA measurements) show some promise as benchmarks of specific modeled microbial processes, but at this stage require more work to determine the factors that control the regulation of mRNA. Previous

work has shown a lack of correlation between the production of mRNA and the activity of the pathway encoded by that mRNA. Post transcriptional modification pathways play and important role in determining the balance between transcription and translation. More specific incubation experiments around the capillary fringe (for example, the use of random isotope pairing techniques to differentiate anammox from denitrification), would be very useful for parsing out metabolisms of importance, however, were beyond the scope of the current study.

Nonetheless, we believe benchmarks for microbial models are an important area to highlight in this manuscript, and have included a section in the discussion that explicitly deals with the benchmarking needs for models of this type.

We disagree, however, with the reviewer's assertion that the model performs poorly in failing to capture the nitrate dynamics. The model does not capture the totality of the nitrate dynamics in the current configuration. This is because the model is being run to examine the extent of biological nitrogen loss from the different depths. We make this point in the materials & methods (Pg. 7 Ln 7 - 8), the results (e.g., Pg. 10, Ln 8 - 11) and discussion. From this perspective, comparison with the Rayleigh calculations from the isotopic data, the model actually performs reasonably in capturing the nitrate dynamics as catalyzed by different microbiological guilds and as a function of the oxygen dynamics, and organic matter/ nitrate concentrations. It is quite possible to configure the model to account for all of the nitrate loss from biological dynamics (as shown in the supplemental figure 4), or under variable electron donor ratios (supplemental figure 5), however, the broad conclusions from the isotopic data suggests that this would be incorrect, and again, highlights the utility of using isotope data to benchmark this model. It is possible that this point is not made clear enough in the current text. Therefore we have added additional text to the discussion to emphasize this point. For comparison, we have also run the model to simulate both abiotic (dilution) and biotic pathways. These simulations are given in the supplemental figures and discussed further in the text. Finally, in order to compare how well the model captures the data, we have run

statistical tests represented in a Taylor Diagram also included in the supplemental, and further discussed in the text.
* * *
The microbial simulations come off as entirely speculative given that there are no data presented, as does the speculation as to the importance of annamox vs. canonical denitrification vs. chemolithotrophic processes. Contrary to the conclusion (P17 25), I don't think the authors can make any concrete claims as to the mechanisms driving the patterns observed, especially given that the nitrate isotope fractions are not well constrained for these pathways, and that there is enormous variation in nitrate isotope fractionation during denitrification.

Our conclusions are drawn predominantly from the simulations, and the conditions under which these simulations are performed. With regards to understanding the importance of nitrogen loss via heterotrophic denitrification Vs. chemolithoautotrophic anaerobic ammonia oxidation, this question is driven primarily by recent molecular microbiology work at this site showing a relatively high abundance of chemolithoautotrophic metabolisms in the groundwater (Jewell et al., ISME, 2016; Frontiers in Microbiology, 2016), and high abundance of ammonia-oxidizing archaea (Hug et al., Envion. Micro, 2015) and heterotrophic denitrifiers (Anantharamam et al., Nat. Comm. 2016) at shallower depths. We do point out in the text that the spin-up conditions (i.e., a low water table fostering aerobic conditions) prior to the water table perturbation simulations can select against obligate anaerobes (such as the anammox bacteria), and for faculative aerobe such as the heterotrophic bacteria.

From this perspective, we do not believe that our interpretation of the model simulations is speculative. The development of the model is informed by prior studies at the same site (Pg. 2 Ln 25), the model parameters are taken from literature values of representative organisms (aerobic and anaerobic ammonia oxidizers & faculative heterotrophs, see supplemental tables), the broad conclusions of the model simulations (i.e., the % of

biotic N-loss Vs. abiotic dilution) are supported by isotopic benchmarks (from Rayleigh fractionation calculations, Pg. 10 Ln 10, and simple mixing calculations, see supplemental material) , and the final question, as to the general importance of anammox Vs. denitrification to N-loss, is supported by prior ecophysiological data and mechanistic modeling. The discussion also goes into more details as to the broader conclusions (i.e., from Pg. 15. Ln 20 onwards) we make from the study. We have, however, added additional text to make it clear that these conclusions are based primarily on model simulations.

The isotopic data has not been used to attempt to parse between the two different pathways. As with previous studies examining the contributions of anammox vs denitrification to nitrogen loss (e.g., Babbin et al., Science, 2014; Koeve & Kahler, Biogeosciences, 2010), we've employed a mechanistic model. As with previous models, it is a simplification of real-world conditions, yet captures some of the more important traits related to fitness under fluctuating environments. Hence, the output is therefore theoretical, rather than speculative, yet corresponds to findings of previous studies attempting to parse out the factors determining nitrogen loss from discrete end members.

We believe that this study therefore supplies suitable impetus for follow up experimental work based on the model output. Furthermore, modifications to the baseline model presented here (for example, incorporating dynamic energy budgets based on the thermodynamic approach explained in the text) could be used to examine why there is such variability in isotopic fractionation from an ecological and metabolic perspective.

———————————————————————————————————————————————

The spatial replication of the field data seems inadequate given the heterogeneity of the system under study. Why are no isotope measurements from the vadose zone and shallow soils reported? This seems critical to get at the question of biogeochemical processing of N vs. dilution or mixing that comes up throughout the paper, and the enormous spatial heterogeneity of nitrate isotope compositions that is increasingly

documented in the literature. What is the composition of the water that is posited to be diluting the sediment zone of interest? There was almost no discussion of the hydrology of the site and potential source waters, which are critical for getting at this point. To interpret the isotopes, you would need to consider mixing rather than pure dilution unless you could demonstrate that you were mixing nitrate-rich vs. nitrate-free water. This is especially critical in the context of the heterogeneity in buried organic lenses that has been demonstrated at this site.

Nitrate accumulates and dissipates only in the depths currently under investigation (i.e., 2 - 3 m below surface depth), with little evidence from this study or from previous studies that nitrate accumulates at shallower or deeper depths. Measurements of nitrate in the vadose zone were below detection (figure 2), it is also unlikely, given infiltration rates at this site ($\sim$ 3 cm yr-1, Pg. 14, Ln 2), that nitrate from shallower soils are transported to = 2 m and below. This is further supported by recent work at the site adding $\sim$ 2500 gallons of deuterium-enriched snow ($\delta$D $\sim$ 2200 per mil), for the purpose of examine water infiltration into the vadose zone around the well used in the current study. Snowmelt last 6 days and $\delta$D rapidly infiltrated to $\sim 1 - 1.5$ m, with very little deuterium signal seen below 1.5 m. Therefore, the transportation of nitrate from the vadose zone to the capillary fringe was not considered to be of importance in the current study. Similarly, nitrate below the 3 m line has been shown to be very low. Fig. 2 shows nitrate data for 3.14 m below surface depth, the lower bound of the current data set, with nitrate concentration ranging from 60 to 700 micromoles. Below this, into the background aquifer, nitrate ranges from undetectable up to 80 micromoles, as reported in previous studies (Zachara et al, J. Cont. Hydrol. 2013; Yabusaki et al., ES&T, 2017). This is alluded to in the main text (Pg. 3, Ln 14), however, we have rewritten this statement to make it clearer. Finally, and further emphasizing the nitrate-deficient conditions in the groundwater, a recent NO3 injection experiment injected $\sim$ 2 mM of nitrate into the groundwater intending to stimulate chemolithoautotrophic metabolisms (Jewell et al., ISME, 2016; Frontiers in Microbiology, 2016). Prior to the injection, nitrate concentrations ranged from undetectable to $\sim$ 70 $\mu$M. Post-injection, the nitrate

was entirely consumed within the first 1 m downgradient.

In summary, the reason that no isotope measurements were made in the vadose zone or background aquifer was that nitrate was often below detection limits of the technique. This would also minimize the likelihood of nitrate from outside the depths studied contributing significantly to the observations reported here.
* * *
I am very surprised that the authors report nitrate concentrations of 5 mM surely they must mean 5 micromolar or 5 mg NO3- L-1?

The mM units are correct. Nitrate is measured routinely at this site by ion chromatography according to approaches reported in the main text (Pg. 4, Ln 10 -11). Data from previous years also shows the large accumulation of nitrate (to mM concentrations) in the unsaturated zone pore water are a recurring phenomenon. An explanation for such high nitrate concentrations is the presence of a natural reduced zone around this well (as discussed on Pg. 14, Ln 11). Organic matter concentrations are very high in these zones, Janot et al., ES&T, 2016, recorded organic matter in these regions with OC concentrations as high as 1.7 %. We can therefore use a back-of-theenvelope calculation to estimate nitrogen availability from the OM in these regions. Considering a measured C:N ratio for the relevant depths of 7 (Conrad et al., unpublished) and a bulk density of $\sim$ 2 g cm-3, OM in these naturally reduced regions could yield 0.004 g-N cm-3, or 290 mM of nitrogen. Using a conservative mineralization rate of 2 % per year would therefore yield $\sim$ 6 mM of nitrogen.

This high nitrogen yield therefore makes this site an excellent candidate to study biological hotspots of activity.
* * *
The manuscript is riddled with errors. In the title alone there is a grammatical error and a misspelling of one of the author names. I urge the lead author to give the paper a

proper proof reading before sending out for review!

The manuscript has been proofread again. However, I (the lead-author) am unable to identify the spelling mistake. Looking at both the file for submission, and the file uploaded to the Biogeosciences-Discussions website, all authors names are spelt correctly.
* * *
For example, P8 line 18, 15N is given as -1.8. At line 23, this same value is referred to but the minus sign is missing. This oversight has been fixed.
* * *
There are many more examples... P4 30: "Samples of pore gas from 2 m bsd" do you mean below 2 m? The abbreviation bsd stands for below surface depth, and is defined on page 3, line 31.

How many depths were sampled? Seven depths were sampled (0.5, 1.0, 1.5, 2.0, 2.5, 3.0, 3.14), and this information can be found in the first paragraph of the materials and methods (Pg. 4, Ln 2).

P9 15: what do you mean by "highest (most reduced) value" This simply refers to the highest enrichment recorded, however, might be confusing, therefore has been reworded.

The message in Fig. 3 is not at all clear as presented. Try putting the same values (15N, e.g.) on a common plot so we can compare the trends among depths over time. We are unsure as to what is unclear here. The figure shows the corresponding enrichment in the 15N18O-nitrate accompanied by the trajectory in nitrate concentration over time. The left yaxis represents the isotopic composition of nitrate (15N/ 18O), and is the same axes (from - 10 to +20 per mil) across all three plots, while the right y-axis is the concentration of nitrate from 0 - 8 mM, and again, is the same axis across all three plots. We are therefore not sure as to the value of re-plotting these figures by 15N.

P13 20: Need citation A citation has been added

P15 25: "the measured in N2O peak" This has been reworded.

Please also note the supplement to this comment:
https://www.biogeosciences-discuss.net/bg-2017-212/bg-2017-212-AC1-
supplement.pdf

---

## Author Comment (AC2) · 12 Sep 2017

Validation and statistical assessment. However, the article in its present form lacks validation through microbial isotope experiments, which would truly provide "evidence" for microbial mediated nitrate cycling.

We have added additional statistical analysis to the current work through the use of a Taylor diagram (Fig. S6). With respect to validation, the model has been tested against several different scenarios related to dilution + denitrification (added into the supplemental, Fig. S7), denitrification alone (Fig. 4 of the main text), increasing OM concentrations (Fig. S4), and varying ratios of electron donors (Fig. S5). The model captures

the changing nitrate concentrations attributable to abiotic and biotic processes, but can also be used to solely capture the nitrate turnover attributable to denitrification (parsing this out from that attributable to dilution). Increasing the organic matter concentration shows a clear correspondence between denitrification and the decline in nitrate, however, the isotopic data suggests that denitrification is responsible for much lower nitrate reduction loss than this.
* * *
The details of that season (e.g., rather it was a normal rainfall year) were not presented.

Further information on precipitation has been added to the materials and methods. The year in which this study was conducted, 2014, had a high snowpack, which drove the water table higher than previous years. Normally the water table does not immerse 2 m bsd, however, regularly saturates the 2.5 and 3 m bsd. This information has been added to the materials and methods section.
* * *
Likely, a map of the monitoring locations would be helpful within the floodplain.

A map of the site showing the location of the study well has been added to the supplemental (Figure S1).
* * *
Further information is needed regarding the soil and water chemistry (e.g. pH), which will impact microbial community population and productivity.

Further information on pH, electrical conductivity and dissolved oxygen has been added to the results section.
* * *
The high nitrate concentrations were surprising - Is this area in an agricultural landscape?

This area is not impacted by agriculture, however, has a number of heterogeneously distributed naturally reduced zones (NRZs). This aspect of the site is discussed on Pg. 14, Ln 11. The presence of these NRZs (which are essentially buried horizons formed by river overbanking), generally explains the high nitrate concentrations. Organic matter concentrations are very high in these regions, Janot et al., ES&T, 2016, recorded organic matter in these regions with OC concentrations as high as 1.7 %. We can therefore use a back-of-the-envelope calculation to estimate nitrogen availability from the OM in these regions. Considering a measured C:N ratio for the relevant depths of 7 (Conrad et al., unpublished) and a bulk density of $\sim$ 2 g cm-3, OM in these naturally reduced regions could yield 0.004 g-N cm-3, or 290 mM of nitrogen. Using a conservative mineralization rate of 2 % per year would therefore yield $\sim$ 6 mM of nitrogen.

————————————————————————————————————————————

"Evidence for microbial mediated nitrate cycling..." seems misleading, assuming no microbial isotopic data was collected in the soils.

The measurements reported by the current manuscript are taken from porewater samples collected under both saturated and unsaturated conditions over the course of the year. The wells drilled into the floodplain site incorporate a suction lysimeter at each depth sampled (as outlined on Pg. 4 Ln. 1), allowing for sampling across the year.

————————————————————————————————————————————

Additionally, do the authors have isotopic data for the confined aquifer to confirm that the mixing water is truly nitrate free?

Nitrate accumulates and dissipates only in the depths currently under investigation (i.e., 2 - 3 m below surface depth), with little evidence from this study or from previous studies that nitrate accumulates at shallower or deeper depths. Measurements of nitrate in the

vadose zone were below detection (figure 2), it is also unlikely, given infiltration rates at this site ($\sim$ 3 cm yr-1, Pg. 14, Ln 2), that nitrate from shallower soils are transported to = 2 m and below. . This is further supported by recent work at the site adding $\sim$ 2500 gallons of deuterium-enriched snow ($\delta$D $\sim$ 2200 per mil), for the purpose of examine water infiltration into the vadose zone around the well used in the current study. Snowmelt last 6 days and $\delta$D rapidly infiltrated to $\sim$ 1 − 1.5 m, with very little deuterium signal seen below 1.5 m. Therefore, the transportation of nitrate from the vadose zone to the capillary fringe was not considered to be of importance in the current study. Similarly, nitrate below the 3 m line has been shown to be very low. Fig. 2 shows nitrate data for 3.14 m below surface depth, the lower bound of the current data set, with nitrate concentration ranging from 60 to 700 micromoles. Below this, into the background aquifer, nitrate ranges from undetectable up to 80 micromoles, as reported in previous studies (Zachara et al, J. Cont. Hydrol. 2013; Yabusaki et al., ES&T, 2017). This is alluded to in the main text (Pg. 3, Ln 14), however, we have rewritten this statement to make it clearer.

Finally, and further emphasizing the nitrate-deficient conditions in the groundwater, a recent nitrate injection experiment injected $\sim$ 2 mM of nitrate into the groundwater intending to stimulate chemolithoautotrophic metabolisms (Jewell et al., ISME, 2016; Frontiers in Microbiology, 2016). Prior to the injection, nitrate concentrations ranged from undetectable to $\sim$ 70 $\mu$M. Post-injection, the nitrate was entirely consumed within the first 1 m downgradient.

In summary, the reason that no isotope measurements were made in the vadose zone or background aquifer was that nitrate was often below detection limits of the technique. This would also minimize the likelihood of nitrate from outside the depths studied contributing significantly to the observations reported here.
* * *
Based on purely visual observations, the model predications do not appear to fit well

with the observed data

The model (as shown in Fig. 4) is representing only biological nitrate loss (i.e., attributable to denitrification or anammox). One of the points that we wish to make with this manuscript is that a model can be forced to replicate data (Figure S3 is an example of this), however, the right results can emerge for the wrong reasons. This emphasizes the importance of using the correct data to benchmark the model. In the current study, simulations presented in figure 4 reflect the qualitative conclusions of the isotopic data (i.e., denitrification is not, and cannot, be responsible for all nitrate loss). These simulations show relatively poor correlation to the observations (as shown through statistical comparisons), yet are valuable for making the point that the correct benchmarks are important.

We have added an additional paragraph to the conclusions reflecting this point. However, additional simulations are included in the supplemental material showing the comparison between observations and simulations when dilution is incorporated at the same time. As expected, this set of simulations shows excellent correlation to the observations. However, the earlier point stands, that setting up the model to simulate in well N-species, and OM concentrations can replicate the extent of nitrate reduction as informed by isotopic data. This is a significant advantage of the current model.
* * *
The second sentence in section 4.1 needs a reference. A reference has been provided.
* * *
Additionally, several of the graphs are difficult to read (e.g. Fig 4, S3, S4). We acknowledge that these figures are complicated, and attempts have been made to improve the contrast within the figures.

Please also note the supplement to this comment:

https://www.biogeosciences-discuss.net/bg-2017-212/bg-2017-212-AC2-supplement.pdf